# Tetrodotoxin and the Geographic Distribution of the Blue-Lined Octopus *Hapalochlaena fasciata* on the Korean Coast

**DOI:** 10.3390/toxins15040279

**Published:** 2023-04-11

**Authors:** Ji-Hoe Kim, Dong-Wook Kim, Sung-Rae Cho, Ka-Jeong Lee, Jong-Soo Mok

**Affiliations:** 1Research & Development Planning and Coordination Department, National Institute of Fisheries Science, Busan 46083, Republic of Korea; 2Food Safety and Processing Research Division, National Institute of Fisheries Science, Busan 46083, Republic of Korea; 3Southeast Sea Fisheries Research Institute, National Institute of Fisheries Science, Tongyeong 53085, Republic of Korea

**Keywords:** blue-lined octopus, tetrodotoxin, geographic distribution, *Hapalochlaena fasciata*, Korea

## Abstract

The genus *Hapalochlaena,* including the blue-lined octopus *Hapalochlaena fasciata (H. fasciata),* is highly toxic. Venomous, blue-lined octopuses were recently found in Korea, but their toxicity, toxin composition, and distribution remain largely unknown. Here we estimated the geographic distribution of the organisms along the Korean coast and clarified their toxicity. Tetrodotoxin (TTX) was present in all three specimens of *H. fasciata* examined, although the toxicity varied largely between individuals. The mean TTX concentration in the whole body of the three specimens was 6.5 ± 2.2 μg/g (range 3.3–8.5 μg/g). Among the body parts examined, the salivary glands exhibited the highest concentration (22.4 ± 9.7 μg/g). From 2012 to 2021, 26 individuals were obtained nearly every month from different regions of the Korean coast. A non-fatal case of a blue-lined octopus bite was reported along the Korean coast in June 2015. This is the first report on the widespread distribution of blue-lined octopuses on the Korean coast and TTX detection. The widespread distribution of the TTX-bearing *H. fasciata* along the Korean coast within the temperate zone indicates that the species may soon become a serious health issue in Korea. The toxicity of this species is also a potentially significant human health risk.

## 1. Introduction

Blue-ringed octopuses are cephalopods belonging to the genus *Hapalochlaena*. Some of these octopuses are highly venomous and poisonous species characterized by small body size and iridescent blue markings on their dorsal surfaces and arms. The genus *Hapalochlaena* is currently considered to include four closely related species. Among them, three species, *H. marculosa*, *H. lunulata*, and *H. fasciata*, are known to be highly toxic; however, little is known about the toxicity of *H. nierstraszi* [1]. These species are widely disseminated in tropical and subtropical reef waters. *H. fasciata*, which is also called the blue-lined octopus, is one of several species of blue-ringed octopus. The species is commonly found in the waters off the coast of Australia, but its distribution extends through the Pacific Ocean, north to Korea and Japan [2,3]. The potent neurotoxin tetrodotoxin (TTX) is found in various tissues of *H. fasciata*, including the salivary glands, muscles, and skin [4,5,6]. It is well-known that TTX is one of the most powerful marine biotoxins [7]. The European Food Safety Authority (EFSA) reported that the safety limit for toxic effects in humans is 44 μg/kg for TTX and its analogs, based on the consumption of 400 g of shellfish meat [8]. The toxin present in the genus *Hapalochlaena*, including *H. fasciata,* is venomous if injected via a bite or poisonous if the organisms are ingested by a predator. A bite from *H. fasciata* or ingestion of the organism is a serious health risk to humans and may be fatal. Several case reports of human bites by the blue-ringed octopus in Australia have been published [9,10,11,12]. *H. fasciata* is a non-commercial species, but this species is responsible for at least one human death [1]. A food poisoning incident due to the ingestion of TTX-bearing *H. fasciata* occurred in Taiwan in 2010 [13]. In Australia, *H. fasciata* is responsible for the fatal envenomation of adult green sea turtles that accidentally ingest the octopuses along with seagrass [14].

Along the Korean coast, a blue-lined octopus similar to *H. fasciata* was found for the first time in the waters off Jeju Island in 2003 [3]. *H. fasciata* was subsequently added as a new species on the National List of Species of Korea [15]. Recently, an individual collected from the coast of Jeju Island in Korea in 2015 was confirmed to be *H. fasciata* on the basis of detailed morphologic observations and molecular analysis [16], and the complete mitochondrial genome of this species collected from the southern coast of Korea in 2019 was identified [17]. Kim et al. [3] reported the spread of the octopus along the Japanese coast side in the East/Japan Sea due to sea warming and suggested its presence along the Korean coast, although there has been no evidence of this except at Jeju Island, located off the southern tip of the Korean Peninsula. Since 2012, there have been 26 reports in news magazines of *H. fasciata* caught from coastal waters of both Jeju Island and the Korean mainland, although the Korean coast is within the temperate zone. The news also reported a suspected case of a blue-lined octopus bite in an adult male.

Information on the toxicity and spread of *H. fasciata* in a newly discovered region is useful in public health. Despite finding these venomous organisms in Korea in recent years, the distribution of *H. fasciata* and its toxicity and toxin composition remains largely unknown. This study aimed to estimate the geographic distribution of the blue-lined octopuses found along the Korean coast and clarify their toxicity and anatomical distribution of TTX.

## 2. Results and Discussion

### 2.1. Anatomic Distribution of TTX in Blue-Lined Octopuses

The selected reaction monitoring (SRM) chromatograms of liquid chromatography-tandem mass spectroscopy (LC-MS/MS) at *m*/*z* 320 are shown in Figure 1. The tissue samples of blue-lined octopuses yielded a chromatogram with a peak at the same retention time (3.92 min) found in a TTX standard and an unknown peak at 3.49 min. Both TTX and two TTX analogs (4-*epi*TTX and 6-*epi*TTX) had the same molecular weight (319 Da) [7]. Especially, TTX exists in chemical equilibrium with 4-*epi*TTX and 6-*epi*TTX [8]. Unfortunately, the TTX analogs were not quantitated because their standards were not commercially available for the present study. Therefore, in this study, one of the unknown peaks was assumed to be 4-*epi*TTX from the literature [18,19] and the analyzed MS response. However, a putative peak for 6-*epi*TTX could not be detected. The concentration of 4-*epi*TTX was calculated following the procedure of Bene et al. [19], using the TTX standard as a reference peak and the relative potency (0.16) of 4-*epi*TTX as compared to TTX [8]. Table 1 shows the concentrations and anatomic distributions of TTX and 4-*epi*TTX in each tissue of three blue-lined octopuses *H. fasciata* collected along the Korean coast: head parts (including head, eyes, and mantle), arms, salivary gland parts (including salivary gland, beak, and hepatopancreas), and other parts (including gonads, heart, kidneys, etc.) using LC-MS/MS. TTX was found in all three blue-lined octopus specimens tested in this study, irrespective of the collection date and region, although the toxicity of the three specimens varied widely. The concentrations of TTX and 4-*epi*TTX in the whole bodies of the three specimens ranged from 3.3 to 8.5 μg/g (mean 6.5 ± 2.2 μg/g) and 0.4 to 1.1 μg/g (mean 0.7 ± 0.3 μg/g), respectively. The TTX level was approximately nine times higher than that of the 4-*epi*TTX in the whole body of *H. fasciata*. The presence of TTX was also identified in two blue-lined octopuses collected in each of two large green sea turtles in Moreton Bay, Australia, using LC-MS/MS, but no toxicity was quantified [12]. In addition, in other studies, TTX was detected in all specimens of *H. fasciata* tested; four were collected in Australia [4], and thirteen were collected in Japan [6].

Among all the body parts of the blue-lined octopuses tested in the present study, the salivary gland parts showed the highest toxicity of TTX with a mean concentration of 22.4 ± 9.7 μg/g, followed by the arms (5.1 ± 1.6 μg/g), the head parts (3.6 ± 0.5 μg/g), and other parts (1.2 ± 1.2 μg/g) (Table 1). The total amount of TTX per each tissue of the octopuses, however, presented predominantly in the arms (45.1 ± 20.8 μg), followed by the salivary gland parts (32.3 ± 20.9 μg), the head parts (7.8 ± 2.5 μg), and other parts (0.6 ± 0.6 μg). These findings demonstrate that of the blue-lined octopus *H. fasciata* collected on the Korean coast, the TTX (approximately 88%) was distributed mainly in the arms (54.1 ± 3.8%) and the salivary glands parts (34.0 ± 7.2%), although TTX was found in all the body parts tested in this study. Similarly, when the toxicity of different body parts was examined using specimens of the greater blue-ringed octopus *H. lunulata* collected in Japan, TTX was found to be present in the salivary gland parts as well as in all of the body parts tested, although the posterior salivary glands exhibited the highest toxicity [7]. In contrast, in another study [20], the total amount of toxin in *H. maculosa* collected in the Philippines presented mainly in the other soft body parts (approximately 76.4%), excluding the posterior salivary glands.

Thus, the present findings revealed that all the blue-lined octopus *H. fasciata* specimens collected on the Korean coast had TTX, with wide inter-specimen variations in toxicity. Larger individuals had higher total amounts of TTX. Other studies have confirmed large individual variations in toxicity within the same species of toxin-bearing octopuses [6,7]. Asakawa et al. [7] reported that the level of TTX in toxic octopuses is related to some environmental factors or comes from food. Therefore, along the Korean coast, further research is needed to investigate the location-, size-, sex-, and tissue-dependent variations in the toxicity of the blue-lined octopuses.

### 2.2. Distribution of Blue-Lined Octopus on the Korean Coast

The place and date when each blue-lined octopus was caught on the Korean coast are shown in Table 2. The locations were marked on a map, and their geographic distribution is shown in Figure 2. The numbers in Figure 2 correspond to the place numbers in Table 2. A total of 26 blue-lined octopuses caught from 17 different regions on the Korean coast were recorded from press releases and newspapers during 2012–2021. Thirteen organisms were from the coast of Juju Island, and the remaining 13 organisms were from the coast of the Korean peninsula mainland. On the coast of the Korean mainland, blue-lined octopuses were caught from Ulsan to Yeosu, located along the southeast coast of the Korean peninsula, which is affected by a branch of the Kuroshio warm current. This is the first report on the widespread distribution of blue-lined octopuses on the Korean coast. Our previous study reviewed the distribution of blue-lined octopuses along the Korean and Japanese coasts in the East/Japan Sea and neighboring waters [3]. Only one of *H. fasciata* on the coast of Jeju Island in Korea was reported until 2010, but the octopuses were reportedly widely distributed up to that point along the Japanese coast. Recently, a similar geographic distribution on the basis of live specimen collection from Japanese coasts was also reported by Yamate et al. [21]. These previous studies demonstrated that the blue-lined octopus recently shifted its distribution northward within the Japanese coast, but there were no observations in Korea. On the Korean coast, the northward expansion of the distribution area of the blue-lined octopus was successfully confirmed in this study.

The number of *H. fasciata* caught each month of the year off the coast of Korea is shown in Figure 3. The blue-lined octopus was caught every month except January, March, and April. The octopus was most often caught along the Korean coast in May and November, with five individuals each month, approximately 19% of the total captured populations (26 individuals). Nevertheless, 12 individuals, approximately 46% of the total catch, were caught in the fall season, from October to December. Although there are actually no data available on the abundance of *Hapalochlaena*, the population densities of the octopuses are expected to exhibit a seasonal variation [22]. In our previous study, we found that the octopus was most frequently observed in the spring season (April) on the Japanese coast in the East/Japan Sea [3]. Yamate et al. [21], however, reported that relatively large numbers of octopuses were captured along the southeast and southwest coasts of Japan during the fall and winter seasons. Thus, the blue-lined octopuses are found throughout the year on the Korean and Japanese coasts, but there was no clear pattern in the seasonal occurrence among the three studies, including the present study. The population density of octopus species is significantly associated with the season, but octopus populations fluctuate widely from year to year with no clear trends. The abundance and distribution of each octopus species are limited by a combination of biotic and abiotic factors [23,24,25,26,27,28]. Further studies are needed to clarify the biology of *H. fasciata* because relatively little is yet known about the organism.

The obtainers who caught the octopus, on the other hand, were divided into several groups, including those who were employed in the fishery-related industry as well as those who visited the sea occasionally. The groups consisted of researchers, fishermen using traps, commercial divers, recreational divers, recreational anglers, and travelers (Table 3). The catch rate of blue-lined octopuses by researchers, fishermen, and commercial divers was approximately 38% (10/26), and that by recreational divers and anglers, and travelers were approximately 62% (16/26). Thus, the number of octopuses caught by occasional visitors was higher than that obtained by occupational workers. These findings suggest that anyone can accidentally encounter the octopus on the Korean coast and be at risk of a bite. In fact, a non-fatal case of a blue-lined octopus bite was reported for the first time in Korea on 10 June 2015 (No. 3 in Table 2). In this case, a 38-year-old adult male tourist was bitten by a blue-lined octopus at a popular beach on Jeju Island, Korea. He said that a small, blue-lined octopus bit his middle finger when he tried to observe it with his bare hands. He experienced paralysis in the injured finger; finger pain and dizziness continued for several days after receiving emergency treatment.

Korean, blue-lined octopuses contain high levels of TTX in their salivary glands and soft tissues (Table 2). TTX selectively binds to voltage-gated sodium channels in motor nerves, causing skeletal muscle paralysis and respiratory failure, followed by death [29]. A bite by the *Hapalochlaena* octopus can be fatal within minutes if not treated appropriately. There have been at least three recorded deaths attributed to *Hapalochlaena sp.* bites, two in Australia and one in Singapore; many more people have been bitten and survived without complications [11,12].

To enhance safety and prevent accidental bites from the blue-lined octopus in new habitat areas, such as the southeast coast of the Korean and Japanese coasts on the East/Japan Sea, it is important to raise awareness and education about this species. Warning signs posted on popular beaches by public health authorities can be helpful for keeping beachgoers safe from the harmful octopus, but several studies indicate that beach safety signs are not as effective as assumed [30,31]. Authorities must take additional reasonable steps to raise public awareness with leaflets, brochures, social media posts, public service announcements, and television advertisements [32]. It is important for beachgoers to know that the blue-lined octopus is a dangerous animal and never to attempt to pick up or handle it with bare hands.

## 3. Conclusions

The blue-lined octopus *H. fasciata* possesses TTX in their various tissues, including the salivary glands, muscle, and skin. To the best of our knowledge, this is the first report on TTX detection and the widespread distribution of *H. fasciata* along the Korean coast. The findings of the present study indicated that all three blue-lined octopuses tested contained TTX with wide inter-individual variations in their toxicity. Among all the body parts of the specimens tested, the salivary gland parts had the highest concentration of TTX (22.4 ± 9.7 μg/g), whereas the total amount of TTX (~88%) was distributed mainly in the arms (54.1 ± 3.8%) and the salivary gland parts (34.0 ± 7.2%). In addition, a total of 26 individuals were obtained from the Korean coast during 2012–2021, and they were caught every month except January, March, and April. The number of octopuses caught by occasional visitors (recreational divers and anglers, and travelers) was higher than that of the octopuses obtained by occupational workers (researchers, fishermen, and divers). In fact, a case of a non-fatal blue-lined octopus bite occurred on the Korean coast in June 2015. These findings suggest that misidentification of the TTX-bearing *H. fasciata* on the Korean coast is a potential risk factor for a human bite, especially if mistaken for an edible octopus. Therefore, it is important for beachgoers to know that these species are dangerous animals and never to attempt to pick up or handle them with bare hands or to eat them.

## 4. Materials and Methods

### 4.1. Standard Toxin and Reagents

For the TTX analysis, the standard TTX toxin (Alomone Labs, Jerusalem, Israel) was used with its free base form. To extract TTX from the tissue samples, analytical-grade acetic acid was purchased from Sigma (St. Louis, MO, USA). Mass spectrometry-grade formic acid (Fluka, Buchs, Germany) and high-performance liquid chromatography (HPLC)-grade methanol (Merck, Darmstadt, Germany) were used to prepare the HPLC mobile phases. Deionized water was obtained from a Milli-Q water purification system (Millipore, Bedford, MA, USA).

### 4.2. Sample Collection

Three specimens of blue-lined octopus *H. fasciata* were caught off the coastal regions of Gijang-gun in Busan City (site 11) and Namhae-gun in Gyeongnam province (site 15) in 2019, located on the southern coast of Korea (Figure 2). The specimens were identified as *H. fasciata* by their morphologic characteristics based on the report by Kim et al. [16]. Figure 4 shows a representative specimen of a live blue-lined octopus collected from Gijan-gun in May 2019. The specimen is characterized by its blue lines on the dorsal and lateral mantle/head, with small brilliant blue rings on the arm crown and arms. The total body lengths of the 3 specimens from the mantle apex to the arm tips ranged from 8.2–11.5 cm, and the wet weights of the 3 specimens ranged from 9.0–16.1 g (Table 4). The samples were transported to the laboratory in a cooler.

### 4.3. Sample Preparation

Upon arrival at the laboratory, the blue-lined octopus samples were immediately dissected and separated into the head parts (including head, eyes, and mantle), arms, salivary gland parts (including salivary glands, beak, and hepatopancreas), and other parts (including gonads, heart, kidneys, etc.) of the individual. The separated samples were weighed, homogenized, and stored below −20 °C prior to analysis.

For TTX analysis, the extracted samples were prepared according to the Korean Food Code [33]. Briefly, each homogenized tissue sample was thawed and extracted in 4 volumes of 0.1% acetic acid by boiling for 10 min. The extracts were allowed to cool to room temperature, centrifuged at 5000× *g* for 10 min at 4 °C, and brought up to a 1:4 (weight: volume) ratio with 0.1% acetic acid. An aliquot of the supernatant was passed through a 0.2-µm membrane filter (Adventec, Tokyo, Japan) for TTX analysis using LC-MS/MS.

### 4.4. LC-MS/MS Analysis

All extracted samples were analyzed in triplicate for TTX using LC-MS/MS, according to the method of Lee et al. [34]. TTX was quantified using a triple-quadrupole mass spectrometer (TSQ Quantum Discovery Max; Thermo Electron, San Jose, CA, USA) coupled to an HPLC system. The HPLC unit consisted of a Surveyor MS Pump Plus and Surveyor AS Plus (Thermo Electron, San Jose, CA, USA). Chromatographic separations were performed on a TSK Gel-amide-80 column (250 × 0.2 mm, 5 μm; Tosoh Bioscience, Tokyo, Japan) preceded by a guard column cartridge (4.0 × 0.2 mm; Tosoh Bioscience). Eluents A (0.1% formic acid) and B (methanol) were used for the linear gradient elution, with 10 µL of injection volume. The mass spectrometer was operated with SRM, detecting in positive ionization mode, with the product ions at *m/z* 302, 256, and 162 from the parent ion at *m/z* 320 for quantification of the TTX (Appendix A). The calibration curve was generated with TTX standard solutions (6.25, 12.5, 25, 50, and 100 ng/mL), which were >0.99 of the correlation coefficient (r^2^) with good linearity. The limit of detection (LOD) and the limit of quantitation (LOQ) were determined using a signal-to-noise ratio (S/N = 3) and a concentration of LOD × 3, respectively. LOD and LOQ were 0.01 μg/g and 0.03 μg/g, respectively. The TTX concentrations in the samples are expressed as μg/g wet weight.

### 4.5. Data Collection for the Geographic Distribution Analysis of Blue-Lined Octopus in Korea

Internet search engines were used to find records regarding the blue-lined octopus on online daily news channels, including newspapers or television. Data were also collected from the press releases of government authorities, such as the National Institute of Fisheries Science (NIFS) and the branch of the Korean Coast Guard, and scientific reports. All information was included regardless of the source (i.e., government, press, or scientific reports). Overlapping information was considered as a single case. The individual records were collected discontinuously from July 2012 to August 2021.

## Figures and Tables

**Figure 1 toxins-15-00279-f001:**
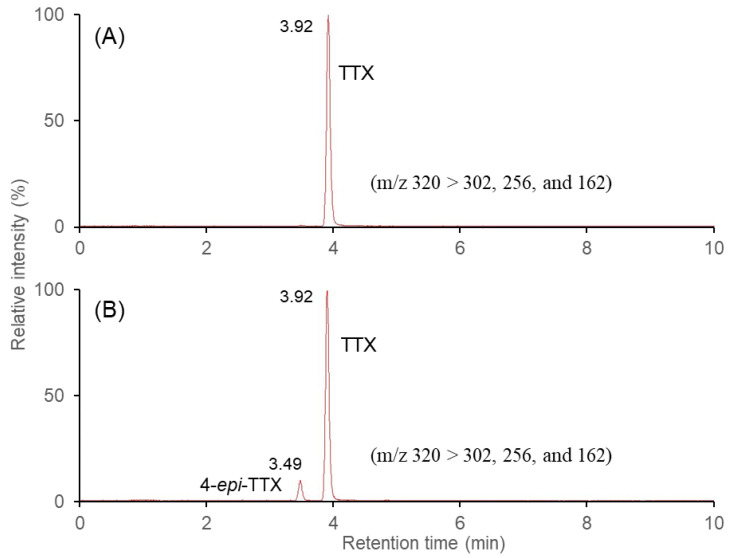
Selected reaction monitoring (SRM) chromatogram of LC-MS/MS obtained from a blue-lined octopus *Hapalochlaena fasciata* collected from the Korean coast. (**A**) 100 ng/mL of TTX standard, (**B**) salivary gland parts of the blue-lined octopus.

**Figure 2 toxins-15-00279-f002:**
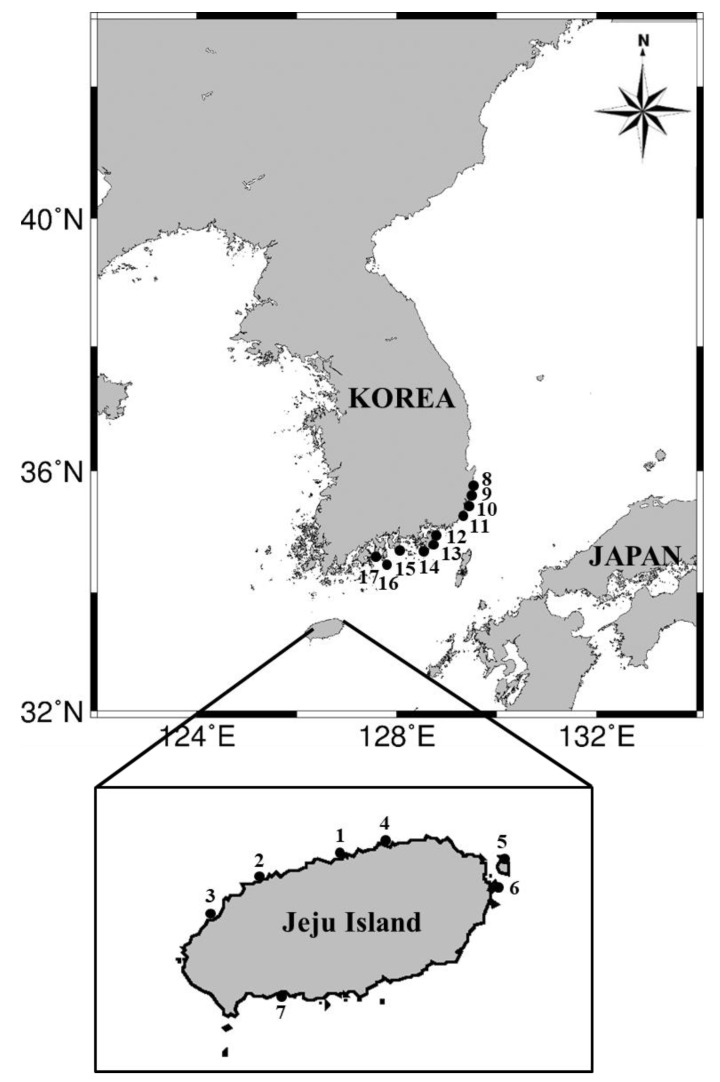
Geographic distribution of the blue-lined octopus caught on the Korean coast. The localities of the numbers correspond to the place numbers in Table 2.

**Figure 3 toxins-15-00279-f003:**
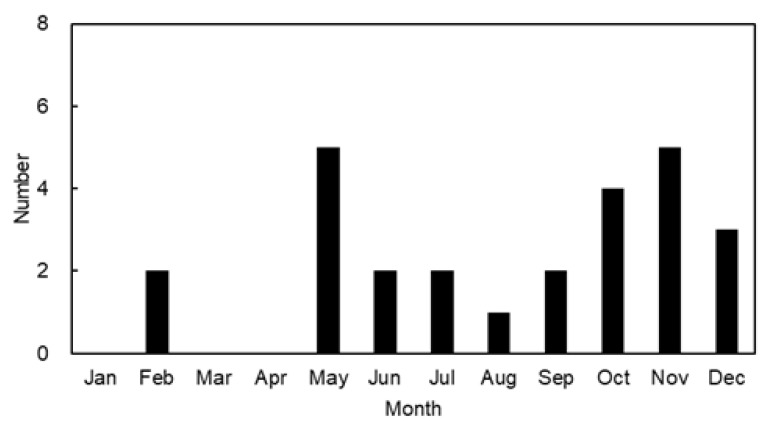
Number of the blue-lined octopus *Hapalochlaena fasciata* caught in each month of the year from the Korean coast during 2012−2021.

**Figure 4 toxins-15-00279-f004:**
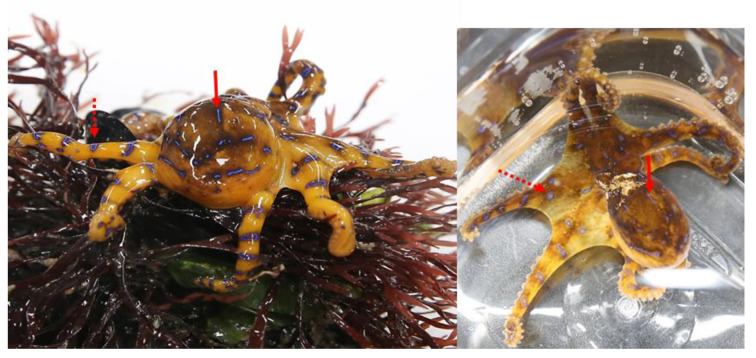
Photograph of a live blue-lined octopus *Hapalochlaena fasciata* collected from the Korean coast. Arrows indicate blue ring arms (dotted line) and the blue lined head (solid line).

**Table 1 toxins-15-00279-t001:** Anatomic distribution of tetrodotoxin (TTX) and 4-*epi*TTX in the blue-lined octopus *Hapalochlaena fasciata* collected from the Korean coast.

Tissue	Collection Site (Date)			
	11 (25 May 2019)	11 (30 July 2019)	15 (4 October 2019)	
	TTX/4-*epi*-TTX (μg/g)			Mean ^(1)^ (μg/g)
Head parts	4.0/0.5	2.9/0.4	3.9/0.4	3.6 ± 0.5/0.4 ± < 0.1
Arms	6.3/1.0	2.8/0.3	6.2/0.7	5.1 ± 1.6/0.7 ± 0.3
Salivary gland parts	34.6/2.9	10.9/0.9	21.8/1.2	22.4 ± 9.7/1.7 ± 0.9
Other parts	0.1/<0.1	0.5/0.1	2.9/0.3	1.2 ± 1.2/0.1 ± 0.1
Whole body	8.5/1.1	3.3/0.4	7.5/0.7	6.5 ± 2.2/0.7 ± 0.3
	Total TTX/4-*epi*-TTX ^(2)^ (μg)			Mean (μg)
Head parts	11.2/1.4	5.2/0.7	7.0/0.7	7.8 ± 2.5/0.9 ± 0.3
Arms	66.8/10.6	17.1/1.8	51.5/5.9	45 ± 20/6.1 ± 3.6
Salivary gland parts	58.8/4.9	7.6/0.6	30.5/1.6	32 ± 20/2.4 ± 1.8
Other parts	0.1/<0.1	0.2/<0.1	1.5/0.2	0.6 ± 0.6/0.1 ± 0.1
Whole body	136.9/16.9	30.1/3.1	90.5/8.4	86 ± 44/9.5 ± 5.7

The tissue samples of blue-lined octopus were divided into 4 parts: head parts (including head, eyes, and mantle), the arms, the salivary gland parts (including salivary glands, beak, and hepatopancreas), and other parts (including gonads, heart, kidneys, etc.). ^(1)^ Mean value ± SD (standard deviation). ^(2)^ Total TTX or 4-*epi*-TTX is represented as the total amount of TTX or 4-*epi*-TTX per each tissue of the blue-lined octopuses.

**Table 2 toxins-15-00279-t002:** List of records of the blue-lined octopus *Hapalochlaena fasciata* collected from the Korean coast: regions and dates of collection.

No. ^(1)^	Obtention Place	Obtainer	Obtention Date ^(2)^	Information Source
1	Jeju-city, Jeju	Researcher	? February 2012	Press Release dated 28 November 2012 by National Institute of Fisheries Science (www.nifs.go.kr (accessed on 22 October 2022))
	Jeju-city, Jeju	Researcher	? November 2012	Press Release dated 28 November 2012 by National Institute of Fisheries Science (www.nifs.go.kr (accessed on 22 October 2022))
	Jeju-city, Jeju	Commercial diver	10 May 2015	Press Release dated 26 May 2015 by National Institute of Fisheries Science (www.nifs.go.kr (accessed on 22 October 2022))
	Jeju-city, Jeju	Recreational diver	? October 2018	The news from 18 October 2018 on the KBS(Korean Broadcasting System) News (www.kbs.co.kr (accessed on 23 October 2022))
2	Aewol, Jeju	Commercial diver	30 May 2014	Press Release dated 22 June 2014 by National Institute of Fisheries Science (www.nifs.go.kr (accessed on 22 October 2022))
	Aewol, Jeju	Recreational angler	10 May 2021	Press Release dated 11 May 2021 by Korea Coast Guard Station Jeju (www.kcg.go.kr/jejucgs/main.do (accessed on 23 October 2022))
3	Hallim, Jeju	Traveler (Bite)	10 June 2015	Press Release dated 29 June 2015 by National Institute of Fisheries Science (www.nifs.go.kr (accessed on 22 October 2022))
	Hallim, Jeju	Traveler	20 September 2021	Press Release dated 21 September 2021 by Korea Coast Guard Station Jeju (www.kcg.go.kr/jejucgs/main.do (accessed on 23 October 2022))
	Hallim, Jeju	Commercial diver	15 November 2021	Press Release dated 15 November 2021 by Korea Coast Guard Station Jeju (www.kcg.go.kr/jejucgs/main.do (accessed on 23 October 2022))
4	Jocheon, Jeju	Recreational angler	9 February 2021	Press Release dated 10 February 2021 by Korea Coast Guard Station Jeju (www.kcg.go.kr/jejucgs/main.do (accessed on 23 October 2022))
5	Udo, Jeju	Recreational angler	1 November 2020	The news from 2 November 2020 on the Jeju mbc News (jejumbc.com (accessed on 23 October 2022))
6	Seongsan, Jeju	Traveler	4 November 2021	Press Release dated 5 November 2021 by Korea Coast Guard Station Seogwipo (www.kcg.go.kr/seoguipocgs/main.do (accessed on 23 October 2022))
7	Seogwipo, Jeju	Researcher	10 September 2015	Kim et al., 2018. [16]
8	Gangdong, Ulsan	Recreational angler	17 July 2017	The news from 20 July 2017 on the Yeonhap News (www.yna.co.kr (accessed on 23 October 2022))
	Gangdong, Ulsan	Fisherman using traps	17 May 2020	Press Release dated 18 May 2020 by Korea Coast Guard Station Ulsan (www.kcg.go.kr/ulsancgs/main.do (accessed on 23 October 2022))
	Gangdong, Ulsan	Recreational angler	18 October 2020	The news from 19 October 2020 on the YTN News (www.ytn.co.kr (accessed on 23 October 2022))
9	Bangeojin, Ulsan	Recreational angler	25 August 2021	The news from 26 August 2021 on the Yeonhap News (www.yna.co.kr (accessed on 23 October 2022))
10	Seosaeng, Ulsan	Recreational angler	18 October 2020	Press Release dated 19 October 2020 by Korea Coast Guard Station Ulsan (www.kcg.go.kr/ulsancgs/main.do (accessed on 23 October 2022))
11	Ilgwang, Gijang	Traveler	25 May 2019	Press Release dated 30 May 2019 by National Institute of Fisheries Science (www.nifs.go.kr (accessed on 22 October 2022))
	Ilgwang, Gijang	Traveler	30 July 2019	The news from 1 August 2019 on the Chosunilbo (www.chosun.com (accessed on 23 October 2022))
12	Deokpo, Geoje	Commercial diver	21 December 2021	The news from 21 December 2021 on the Geojeshinmun (www.geojenews.co.kr (accessed on 23 October 2022))
13	Ilwoon, Geoje	Traveler	5 June 2017	The news from 7 June 2017 on the Kyeongnamshinmun (www.knnews.co.kr (accessed on 23 October 2022))
14	Nambu, Geoje	Fisherman using traps	? December 2021	The news from 1 January 2022 on the Saegeoje News (www.saegeoje.com (accessed on 23 October 2022))
15	Mijo, Namhae	Recreational angler	4 October 2019	The news from 6 October 2019 on th Yeonhap News (www.yna.co.kr (accessed on 23 October 2022))
16	Nam, Yeosu	Recreational angler	15 November 2019	The news from 15 November 2019 on the Yeonhap News (www.yna.co.kr (accessed on 23 October 2022))
17	Hwajeong, Yeoso	Fisherman using traps	3 December 2019	The news from 3 December 2019 on the Yeonhap News (www.yna.co.kr (accessed on 23 October 2022))

^(1)^ The place number is represented as a number in Figure 2. ^(2)^ The unknown obtention date is represented as question mark.

**Table 3 toxins-15-00279-t003:** Number of the blue-lined octopus *Hapalochlaena fasciata* caught by obtainer groups in Korean coastal waters during 2012−2021.

Region	Obtainer						Total
	Researcher	Fisherman Using Traps	Commercial Diver	Recreational Diver	Recreational Angler	Traveler	
Jeju Island	3	0	3	1	3	3	13
Southeastern Coast	0	3	1	0	6	3	13
Total	3	3	4	1	9	6	26

**Table 4 toxins-15-00279-t004:** Sample list for the toxicity analysis of the blue-lined octopus *Hapalochlaena fasciata* collected from the Korean coast.

Collection Site (Date)	Total Length (cm)	Weight (g)				
		Head Parts	Arms	Salivary Gland Parts	Other Parts	Whole Body
11 (25 May 2019)	10.6	2.8	10.6	1.7	1.0	16.1
11 (30 July 2019)	8.2	1.8	6.1	0.7	0.4	9.0
15 (4 October 2019)	11.5	1.8	8.3	1.4	0.5	12.0
Mean ^(1)^	10 ± 1	2.1 ± 0.5	8.3 ± 1.8	1.3 ± 0.4	0.6 ± 0.3	12 ± 3

The tissue samples of blue-lined octopus were divided into 4 parts: the head parts (including head, eyes, and mantle), the arms, the salivary gland parts (including salivary glands, beak, and hepatopancreas), and other parts (including gonads, heart, kidneys, etc.). ^(1)^ Mean value ± SD (standard deviation).

## Data Availability

The authors declare that the data supporting the findings of this study are available within the article.

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
