# Peer review of "Tetrodotoxin and the Geographic Distribution of the Blue-Lined Octopus Hapalochlaena fasciata on the Korean Coast"

_toxins, 2023, doi:10.3390/toxins15040279_

Round 1
Reviewer 1 Report
The manuscript contains interesting information about the distribution of a toxic octopus species in Korea.
The amount of well contrasted scientific information is very limited, but, in any case, it is relevant.
The manuscript, notwithstanding, is not well written and has to be substantially improved.
TTX levels considered to be dangerous should be included in the introduction.
All text, but specially the introduction should be polished and check, preferably, by an English native speaker. The reiterative use of octopus and/octopuses should be avoided.
Introduction
Line 25. Some of these octopuses
Line 28. Spp should be replaced by genus
Line 33 range > distribution
Line 35 organs or tissues
Material and methods
Line 205. Deionized water was passed through… Is this true (in such a case, describe how water was deionized) or was the water obtained from…
Line 234 “solution samples”. Does it refer to extracts?
Even when it seems likely that the detected peak was TTX, separating at least one transition for quantification and another one for confirmation should be done. These traces should be plotted in fig 1.
LOD and LOQ are not given
Results and discussion
Some text describing materials and methods should be removed from the results and discussion section (for example, lines 64-66, 72)
Line 75-79 This sentences are unclear. Reword them.
Line 129 …during the spring through fall seasons. This sentence is unclear.
Figure 1. Caption is unclear. Additionally a peak at 3.49 min is labeled but it is not commented in the text.
Table 2. Obtention instead of obtained in the headers
-Simplify the text of the obtainers (woman diver = diver = recreational diver, for example) Are those differences necessary. If that is the case they should be explained in the text.
Table 3. Should be replaced by a figure.
Table 4. Simplfy header.
Section 2.2 (line 98). Avoid starting the section with the description of tables and figures. That should be included in the figures and tables captions. Start the section with true results.
References
References need to put scientific names in italics and to add doi .
Author Response
We deeply appreciate for the effort and comments of reviewers on our manuscript. We modified and revised the manuscript following the comments, suggestions and recommendations of reviewers.
1. Introduction
→ According to the reviewer comment, we added or modified in Introduction of the revised manuscript.
→ The English in this manuscript has been reviewed by two professional editors, native speakers of English.
2. Material and methods
→ According to the reviewer comment, we added or modified some sentences in the revised manuscript.
3. Results and discussion
→ According to the other reviewer comment, we added or modified or removed some sentences in the revised manuscript.
→ We modified Table 3 to Figure in the revised manuscript.
4. Reference
→ According to the reviewer comment, we reviewed the manuscript. The scientific names were italicized in References in the MS word file of our manuscript. But the scientific names italicized in References were modified during converting to PDF file.
Reviewer 2 Report
1.There is a format error in table 1 and table 5, which should be in a three-line form.
2.It is suggested to delete Table 3, because the information in this table is contained in Table 2.
3.The contents of Total TTX in Table 1 are incomprehensible without detailed description, and the research significance of the experimental results is not explained.
4. With Table 2, when the contents are paginated, the table header needs to be added again.
5. Both the genus and the species names have to be italicized in References. Such as Line 274、Line 277 et al.
Author Response
We deeply appreciate for the effort and comments of reviewer on our manuscript. We modified and revised the manuscript following the comments, suggestions and recommendations of reviewer.
1. There is a format error in table 1 and table 5, which should be in a three-line form.
→ According to the reviewer comment, we modified that the maximum is 2 in the revised manuscript.
2. It is suggested to delete Table 3, because the information in this table is contained in Table
→ According to the other reviewer comment, we modified Table 3 to Figure in the revised manuscript.
3. The contents of Total TTX in Table 1 are incomprehensible without detailed description, and the research significance of the experimental results is not explained.
→ According to the reviewer comment, we added the detailed description of Total TTX. We found that some sentences in the MS word file of our manuscript were missed during converting to PDF file.
4. With Table 2, when the contents are paginated, the table header needs to be added again.
→ According to the reviewer comment, we reviewed the manuscript. The Table 2 were in the same page in the MS word file of our manuscript. But the Table 2 were divided two pages during converting to PDF file.
5. Both the genus and the species names have to be italicized in References. Such as Line274、Line 277 et al.
→ According to the reviewer comment, we reviewed the manuscript. The genus and the species names were italicized in References in the MS word file of our manuscript. But the genus and the species names italicized in References were modified during converting to PDF file.
Reviewer 3 Report
General
This manuscript reports on the findings of blue-ringed octopus in Korean waters. These species of venomous octopus are well known as toxin-produced and are certainly incorporated into public knowledge (along with pufferfish) as being serious risk to public health. With that in mind, however, there are not a huge number of studies describing prevalence of the species that incorporate data on toxin levels (and profiles of toxins) within the tissues of the octopus. As such I certainly welcome this study, as it provides important data on the toxicity of this species. A good number of samples were included in the study, and it is useful data to show toxin distribution within the different tissue compartments within the organisms. My main criticism concerns the lack of information/investigation into the presence of other TTX analogues (which are well known from literature), as not including these in the analysis may result in the significant under-estimation of toxicity. Also Section 2.1 is brief (certainly in comparison to section 2.2) and thus gives the impression of the manuscript being rather imbalanced. If the results and discussion is combined, then more discussion is required for section 2.1. I also found the introduction rather short and brief, so I would like the authors to check and ensure all relevant references are incorporated into the introduction.
Specific
· L9 – use “three” rather than “3”, (spell numbers up to and including twelve, above which numerals are used)
· L12 – use of the word “cases” infers that these are all linked to intoxication events. If this is not the case, then please reword to improve the clarity i.e. that 26 organisms were collected. The term “case” should really just be used for incidents of human health (as used in L13)
· L47-49 – please clarify the wording to state whether this is the same organism referred to in reference 3, or whether this is a separate organism collected later from the same region.
· L54 – can the authors provide any approximate guide to the actual number of reports in the media about octopus occurrence? Are there hundreds, or do the sample numbers in this study represent well the actual numbers found?
· Introduction – general point. Clearly there are well recognised reports of blue-ringed octopus occurrence in tropical/sub-tropical regions of the world. The whole situation with finding these species in Korea would be nice to portray in the introduction about environmental conditions. In particular please can the authors clarify what type of environment the waters are in. Are these temperate (or some other definition?). The aim should state that there is a hypothesis that either the octopus are potentially spreading into temperate regions and/or sea temperatures are increasing in Korean waters, thus resulting in the introduction of invasive species such as this octopus species. This needs to be put forward as a concept in the introduction please (bearing in mind that some international readers of the manuscript may not realise what the marine conditions are like in Korean waters)
· Final paragraph of the introduction – please ensure all aspects of the results/investigation are included in the study aims (currently missing the investigation into anatomical distribution – which is useful and interesting scientific aim to highlight at the beginning).
· Figure 1 – please clarify whether the chromatograms presented are either TICs (summing the three SRMs incorporated into the acquisition method) or are the SRMs for the primary (quantitative) transition (presumably 320 > 302)
· Section 2.1 including Table 1: Throughout the section there is reference to TTX only. However Table 1 presents not just TTX but also “Total TTX”. It is unclear what this refers to specifically and there appears to be no text describing and discussing this. This needs to be clarified and the heading title for Table 1 also needs to explain what the Total TTX refers to very clearly. Presumably this relates to the weight of the tissues rather than a sum of TTX and other TTX analogues – but please clarify
· The toxin results also miss an opportunity to talk about other TTX analogues. TTX standards commonly prepared contain a range of other TTXs, as well as many references in the literature showing evidence for occurrence of these in pufferfish, bivalves gastropods etc. I feel that it is important for the authors to clearly state about the potential for occurrence of other TTX analogues in the octopus samples. It is entirely possible that the total toxin levels may be significantly higher than currently reported if indeed other analogues are present. It should not be much work to look for these analogues and to report their presence or absence. If this work has already been performed and the other analogues not found, then this needs to be stated.
· Also in section 2.1, bearing in mind this is discussion as well as result, the discussion aspect is rather brief and undetailed. My preference would have been to have separate results and discussion sections – but if this is not followed, a greater discussion depth would be required
· Section 2.2 contains some good and interesting data – well presented both with a table and figure showing geographical location. Reading this made me feel as though more was needed in section 2.1 on the science and previous findings, to balance section 2.2
Author Response
We deeply appreciate for the effort and comments of reviewer on our manuscript. We modified and revised the manuscript following the comments, suggestions and recommendations of reviewer.
1. Abstract and Introduction
→ According to the reviewer comment, we modified and added the sentences in the revised manuscript.
2. Results and Discussion: Section 2.1
→ According to the reviewer comment, we reviewed the manuscript, and found that some sentences in the MS word file of our manuscript were missed during converting to PDF file.
→ According to the comment, we added the ratio of the peaks of the standard and tissue sample In the MS SRM analysis as Supplementary Fig. 1 in the revised manuscript.
→ According to the reviewer comment, we reviewed the manuscript including other TTX analogs. The presence of other TTX analogues is significant as the comment of reviewer. But we could not quantitate other TTX analogues because we could not purchase their standard toxins at that time (2019 year). Therefore, we analyzed only the TTX, having the standard toxin, which is a major toxin of TTXs in blue-ringed octopus.
→ In this study, an unknown peak was also detected at 3.49 min except TTX from the SRM chromatograms of LC-MS/MS at m/z 320. An unknown peak was assumed as 4-epiTTX from the literatures and the analyzed MS response and it was described in the revised manuscript.
→ This article presents the first report on the detection of TTX and the widespread distribution of blue-lined octopuses along the Korean coast.
Round 2
Reviewer 1 Report
It seems that the authors are putting a limited effort it the revision of the manuscript. Among other things, they have not included a detailed and comprehensive list explaining the way in which each reviewer comment was addressed.
Line 38 the reference to TTX crystal is not relevant, but some reference about the safety limits for TTX is needed (EFSA. Scientific opinion on the risks for public health related to the presence of tetrodotoxin (TTX) and TTX analogues in marine bivalves and gastropods. EFSA J. 2017, 15, 4752.) or/and others.
As commented in my previous review, the authors should make an effort to reduce the use of the words octopus or octopuses in the text. They are used 9 times in the 18 lines of the abstract and 18 times in the 40 lines of the introduction: a total of 27 times in 58 lines, which gives a poor impression of the document.
Line 70.” … yielded a chromatogram identical to the TTX standard at a retention time of 3.92 min …”. ” … yielded a chromatogram with a peak at the same retention time (3.92 min) than the found in a TTX standard …” . Or a similar sentence
Line 74-75 “Therefore, in this study, an unknown peak was assumed as 4-epiTTX from the literatures [17,18] and the analyzed MS response” à …one of the unknown peaks was assumed to be 4-epiTTX from the literature … Or a similar sentence
Figure 1. The peak at 3.49 and the meaning of the transitions should be explained.
Table 2 header. Correct “Obtetion” à Obtention
Table 3. reformat headers
References. DOI has to be added to the references
Author Response
Title: Tetrodotoxin and the Geographic Distribution of the Blue-lined Octopus Hapalochlaena fasciata on the Korean Coast
We deeply appreciate for the effort and comments of reviewer on our manuscript. We modified and revised the manuscript following the comments and suggestions as attached file.

Reviewer 2 Report
Authors have made subtantial revisions, there are some minor suggestions:
1. In fig. 1, the sentence is not complete "(B) salivary gland 87 parts of the blue-lined";
2. In Tables 1 and 4, no information about "Collection site";
3. In Table 2, revise "Obtetion".
Author Response

(The authors gave the same response as above.)
